# Evaluation of the Pulp Chamber Temperature during Tooth Veneer Preparation Using Burs with Different Degrees of Wear—A Preliminary In Vitro Study

**DOI:** 10.3390/dj11080197

**Published:** 2023-08-15

**Authors:** Edmond Ciora, Mariana Miron, Daliana Bojoga, Diana Lungeanu, Anca Jivanescu

**Affiliations:** 1Department of Oral Rehabilitation and Dental Emergencies, Faculty of Dentistry, “Victor Babes” University of Medicine and Pharmacy, P-ta Eftimie Murgu 2, 300041 Timisoara, Romania; ciora.edmond@umft.ro (E.C.); mocuta.daliana@umft.ro (D.B.); 2Digital and Advanced Technique for Endodontic, Restorative and Prosthetic Treatment TADERP, 300070 Timisoara, Romania; jivanescu.anca@umft.ro; 3Interdisciplinary Research Center for Dental Medical Research, Lasers and Innovative Technologies, 300070 Timisoara, Romania; 4Center for Modeling Biological Systems and Data Analysis, “Victor Babes” University of Medicine and Pharmacy, 300041 Timisoara, Romania; dlungeanu@umft.ro; 5Department of Functional Sciences, “Victor Babes” University of Medicine and Pharmacy, 300041 Timisoara, Romania; 6Department of Prosthodontics, Faculty of Dentistry, Victor Babes, University of Medicine and Pharmacy, Eftimie Murgu Square No. 2, 300041 Timisoara, Romania

**Keywords:** prosthetic tooth preparation, diamond burs, degrees of wear, pulp temperature

## Abstract

The heat produced during tooth preparation could be a source of damage for dental pulp, and many variables are involved in this process. The aim of this in vitro study was to evaluate whether the different degrees of wear of the diamond burs significantly influenced the temperature changes in the pulp chamber during tangential veneer preparation. The sample comprised 30 intact permanent monoradicular teeth, randomly assigned to three study groups of 10 teeth each, of which 5 had the pulp tissue preserved and 5 had thermoconductive paste in the pulp chamber. For prosthetic preparation, we used new burs in the first group, burs at their fifth use in the second group, and burs at their eighth use for the third group. The pulp chamber temperature was evaluated at the start, after one minute, and after three minutes of preparation, using a k-type thermocouple. The results of the three-way ANOVA and Tukey post hoc comparisons showed a highly significant effect of the time of measurement, while the pulp condition and the degree of wear of the burs had no effect. In conclusion, the different degrees of wear of conventional diamond burs do not produce statistically significant different changes in the pulp chamber temperature.

## 1. Introduction

The dental pulp is a specialized organ, unique in the human body, whose primary function is formative, due to the odontoblasts, and the secondary functions relate to tooth sensitivity, nutrition and defence process [1,2]. Irritation of the pulp tissue causes the alteration of odontoblasts, with the appearance of the inflammatory infiltrate, the degree of inflammation being proportional to the intensity and persistence of the irritating factors [1,2]. The etiological factors of pulpal micro-irritation can be microbial, chemical, thermal, and mechanical [3,4,5,6,7,8,9,10,11].

Since 1958, researchers have been concerned with the effect of using the rotary tool at high speeds on the pulp [12,13]. The consequences of increasing the intrapulpal temperature were evaluated by Zach and Cohen in 1965, in a study carried out on the teeth of the Macaca rhesus monkey and they showed that an increase of 5.5 °C at the level of the healthy pulp produced dentin denaturation and destruction of most odontoblasts, with displacement of the nuclei in the dentinal tubules [14]. Even if after two weeks the effects were reversible in 85% of the specimens and the deposition of secondary dentine was observed, the authors concluded that this is the critical temperature threshold for the pulp tissue and the increase of the intrapulpal temperature by over 5.5 °C produces irreversible alterations of the pulp tissue, such as inflammation, abscess and necrosis.

Many researchers have stated that the vital teeth are more impenetrable to bacterial invasion into the dentinal tubules than non-vital teeth and more resistant to fracture than teeth with root canal fillings, thereby suggesting that the vital pulp plays some important role in this process [15,16].

The preservation of pulp vitality following dental intervention is dependent on the degree to which the odontoblasts can survive and initiate an appropriate repair response. So, one key requirement of a successful restorative procedure is to cause minimal additional irritation to the pulp, to not interfere with normal pulpal healing [2,17].

Currently, prosthetic restorative treatments are very widespread, even in young patients. Nonetheless, restorative procedures must lead to aesthetically pleasing outcomes that align with the natural characteristics of natural teeth. The objective is to maintain a maximum amount of dental structure while decreasing the chances of cracks and fractures by adopting the minimally invasive strategy with the least impact on biological integrity, a standard practice in restorative dentistry [16,18].

Moreover, it is well known that the heat produced during restorative procedures is a source of trauma for the dental pulp and has been suggested to lead to inflammation and necrosis [19,20]. This is because during tooth preparation a variable amount of heat is transmitted to the tooth, depending on the type of bur, cutting time, and rate, the pressure applied, the cooling technique, and the speed and torque of the rotary instrument [19,20,21].

Indirect veneers offer a viable substitute to complete coverage restorations, avoiding invasive grinding and excessive reduction of dental tissue from the oral surface, thus ensuring the preservation of tooth structure. Zarow et al. [22] claimed that ceramic veneers exhibited favorable clinical outcomes both on vital teeth (VT) and on non-vital teeth (NVT).

Proper tooth preparation is essential for better aesthetics, acceptable prosthetic rehabilitation, fracture resistance, and healthy soft tissues [23,24,25]. For this purpose, dental burs, fixed to the ultra-high-speed handpieces, are commonly used in dental offices. Regarding the characteristics of diamond burs, cutting efficiency is one of the most important aspects to consider when selecting a diamond bur. In dentistry, the cutting efficiency of diamond burs is generally affected by several factors, including the grit size [20,26,27], the coolant flow [28,29], the load applied by the operator, the design, the tooth structure [20,26], the repeated use [26,27], the cleaning, and the sterilization [26,30,31], the use of the electric micromotor vs air turbine handpieces and the cutting technique [4,29], the number and location of the dental handpieces coolant ports [4,29]. Many studies have stated that the cutting efficiency decreases as the number of cuts increases, regardless of the type of burs, and the reduction is the highest after the first use [23,26,30,31]. Consequently, worn-out burs might necessitate excessive pressure during tooth preparation, leading to undesired frictional heat generation that can potentially induce pulpal inflammation. Intrapulpal temperature changes can be studied either by in vivo or in vitro study. In vivo studies are desirable, but there are several issues associated with this experimental condition; therefore, in vitro studies are widely used for evaluating the intrapulpal temperature changes [4,19,21,23,27,32,33].

The aim of this in vitro study was to evaluate whether the different degrees of wear of the diamond burs significantly different influenced the temperature changes at the level of the pulp chamber, during the tangential preparation of the vestibular tooth surface for ceramic veneers.

Thus, the null hypothesis states that the different degrees of wear of conventional diamond burs used for veneer preparation do not produce statistically significant different changes in the temperature of the pulp chamber of the tooth.

## 2. Materials and Methods

### 2.1. Experiment Design, Sample Selection, and Ethical Aspects

The study had a full factorial design, with the temperature as the variable of interest and the following three factors: (a) the time of measurement during veneer preparation; (b) the degree of the burs’ wear; (c) the dental pulp status.

This in vitro study was conducted within the Discipline of Oral Rehabilitation and Dental Emergencies, Faculty of Dentistry of the Victor Babes University of Medicine and Pharmacy in Timisoara. The research design and protocol received approval from the Research Ethics Committee (approval number 15 of 31 March 2023) of the Victor Babes University of Medicine and Pharmacy in Timisoara. The study included thirty vital permanent monoradicular upper teeth (*n* = 30), extracted due to the loss of periodontal attachment from patients (aged 50–65) selected from the Oral Rehabilitation and Dental Emergencies Discipline patients. The teeth were approximately the same dimensions with no carious lesions or endodontic treatment.

### 2.2. Materials and Study Protocol

After extraction, the teeth were cleaned and stored in distilled water at room temperature (25 °C ± 0.5) for maximum 24 h. Next, on each tooth, a tunnel was created in the middle third, perpendicular to the lingual surface of the tooth, in order to place the thermocouple inside the pulp chamber. For this purpose, we used a Komet^®^ 6801 Round Diamond Preparation Bur (Manufacturer Item Number: 6801.FG.016.A2, Trophagener Weg 25, Lemgo, Germany), with a diameter of 1.6mm. All preparations were performed by the same operator, in the same room where the teeth were stored, maintaining the same temperature of 25 °C throughout the study.

Fifteen teeth were studied under the condition in which the pulp tissue was retained; for the other fifteen teeth, the pulp chamber content was removed using an excavator and thorough cleaning of the pulp chamber was achieved by rinsing with 5.25% sodium hypochlorite followed by saline solution. Later, the emptied cavity was filled with thermoconductive paste (Thermal Grizzly Hydronaut (1 g), Model: TG-H-001-RS, Thermal Grizzly, Hohen Neuendorf, Germany). Then, the teeth were fixed in a silicone mass at the root level by using Zhermack ZetaPlus putty (Zhermack S.p.A., Via Bovazecchino, 100/Badia Polesine (RO) ITALY) high-viscosity condensation silicone. The thermocouple was inserted 1 mm deep into the pulp chamber and immobilized using a heat-free light-cured liquid dam (LC Block-Out Resin, Ultradent, Products GmbH, Am Westhover Berg 30, Cologne, Germany), which was applied surrounding it. In order to have a reproducible position at all moments of data collection, a reference sign was marked 1 mm away from the end of the thermocouple. Subsequently, the selected teeth were randomly assigned to three study groups of 10 teeth each, of which 5 teeth had the pulp tissue retained and 5 teeth had thermoconductive paste in the pulp chamber:Gr. A (*n* = 10): selected to be prepared with new burs (first use);Gr. B (*n* = 10): selected to be prepared with burs at their fifth use;Gr. C (*n* = 10): selected to be prepared with burs at their eighth use (Table 1).

For each test group, the temperature in the pulp chamber was evaluated at three moments of the study: initial, during (after one minute) and following the prosthetic preparation (after three minutes) with the selected burs.

The acquisition of the pulp chamber temperature values is shown in Figure 1 and was carried out with the help of the k-type thermocouple (TC), of 1 mm in diameter, connected to a 2701 Ethernet Multimeter Data Acquisition System, Keithley (USA); the signal was recorded, transformed, and visualized with specific software version 19.0f2 (32 – bit), designed in LabVIEW and installed on the Laptop HP ZBook 15 G3, 15.6″, Intel Core i7-6820HQ, 16GB DDR4, 512GB SSD (Figure 1).

After tooth immobilization, a three-tier diamond depth cutter, with a thickness of 0.5 mm (Meisinger, depth cutter 834, FG 806 314 552 524, L—0.5 mm, Neuss, Germany), was used to cut across the vestibular surface, in order to draw the guide grooves, respecting the axial inclinations (cervical, middle, and incisal) to preserve the convergence of the tooth surface (Figure 2). Then, the vestibular surface was prepared using diamond burs with different degrees of wear for approximately three minutes each, by the same operator and using the same cutting technique. In this way, the depth of tooth preparations was standardized to 1 mm, which allowed for maintaining a consistent barrier (dentin) between the cutting instrument and the pulp chamber at a minimum of 2 mm. Moreover, the thickness of the hard tissue layer was measured before and after the vestibular preparations, using a caliper. Any tooth showing anatomic variation and allowing increased or reduced dentin thickness that could influence the thermal measurement was excluded.

The diamond burs included in the study were used for the first time (new) in group one, for the fifth time in group two, and for the eighth time in group three. In other words, the burs for group two had already been previously used four times for other dental grindings, and those for group three had already been used seven times, at three minutes per use. We chose the minimum threshold of five uses of the bur because, in the specialty literature, it is recommended to not use a bur for grinding more than four times [10]. The burs used for the preparation were from Komet dental, cylindroconical bur 859UF dimensions/sizes: 14; size diameter: 1/10 mm; length: 9.0 mm; maximum speed: 300,000; angle: 3.7°. The handpiece utilized in this study was the TG 656 EASY turbine from Chirana MEDICAL^®^ (Stará Turá, Slovakia) featuring a 5-hole spray for optimum cooling and equipped with a built-in push button system for drill-bit fixation and ceramic bearings. The turbine was connected to the dental unit via a universal turbine hose 4-ISO MIDWEST. The preparations were performed under a water flow rate of 50 mL/min [34] and the cooling water reservoir was filled with water at a temperature of 20 °C ± 0.5 (Figure 3). The recommended operating pressure was from 0.21 to 0.23 MPa, with a maximum of 0.30 MPa and a maximum bur rotation of 300,000 rpm ± 10%. The hygiene of the turbine was secured by sterilization at 135 °C. Data collection was carried out by the same operator.

The burs went through the same cleaning and sterilization protocol (decontamination, mechanical debridement, ultrasonic bath, rinsing, and sterilization (autoclave 135 °C)) until they reached the required number of uses.

The study protocol is shown in Figure 4.

### 2.3. Statistical Analysis

The descriptive statistics included the mean and standard deviation of the temperature for each combination of categorical variables (namely, the three considered factors). To test the statistical significance of the observed variations in temperature, three-way analysis of variance (ANOVA) was applied, followed by post hoc comparisons according to the Tukey procedure. The normality of the values’ distribution was checked with the Kolmogorov–Smirnov statistical test.

The statistical analysis was conducted at a 5% level of statistical significance, and all reported probability values were two-tailed. The analysis was performed using the statistical software IBM SPSS v. 20.0. (Armonk, New York, NY, USA).

## 3. Results

The temperature values measured in this experiment are presented in Table 2 (for each of the 18 five-tooth combinations of factors), and the 95% confidence intervals for the estimated mean values are illustrated in Figure 5.

Table 3 shows the results of the three-way ANOVA: the highly significant effect of the time of measurement can be observed, while the pulp condition had no effect. Post hoc comparisons were conducted applying the Tukey procedure (Table 4).

The results presented in Table 4 displayed a highly statistically significant difference between the temperatures captured at the level of the pulp chamber at the three considered evaluation moments, while the different degrees of wear of the burs had no effect.

## 4. Discussion

The purpose of our study was to evaluate whether the different degrees of wear of the conventional diamond burs could produce statistically significant different changes in the temperature of the pulp chamber, during tangential prosthetic preparation of the vestibular tooth surface.

The values obtained for the intrapulpal temperature, as the variable of interest, were analysed in relation to: the time of measurement during veneer preparation, the degree of the burs’ wear, and the dental pulp status.

The results obtained after the statistical analysis of the collected data showed a decrease in the temperature at the level of the pulp chamber that was highly statistically significant (at a statistical significance threshold of *p* < 0.01), both after 1 min and after 3 min of prosthetic preparations of the vestibular surface of the tooth. The temperature measurements, in the group with pulp tissue retained, indicated a mean temperature drop of Δt = 6.82 °C, after 1 min and a decrease of Δt = 7.82 °C, after 3 min of vestibular prosthetic preparation. For the teeth with the thermoconductive paste in the pulpal chamber, the values recorded after 3 min of prosthetic preparation showed a mean temperature drop of Δt = 8.61 °C. For both study conditions and regardless of the degree of wear of the burs used, the temperature recorded after 3 min of preparation dropped close to 20 °C ± 0.5, that of the cooling water.

There is a consensus in the specialty literature regarding the mandatory use of cooling water during tooth preparation with a high-speed handpiece, as this cooling technique is considered accessible and effective in preventing an intrapulpal temperature increase [27,28].

According to our study results, the greatest reduction in temperature occurred after the first minute of prosthetic preparation, when using water cooling at a flow rate of 50 mL/min and at a temperature of 20 °C ± 0.5. These results are in line with the observations of Lau et al. [4], Galindo et al. [19], Farah et al. [27], Chua et al. [28], and Laforgia et al. [32]. Hence, the temperature and the flow rate of the water coolant are considered the main variables which influence the reduction of the heat transfer to the tooth [4,27,28,33]. These results are consistent with those of Thomas et al. [31], who demonstrated that the use of room temperature water cooling led to a decrease in the pulp temperature regardless of the residual dentin thickness. In our study, the operator used the same cutting technique, intermittently, and the remaining dentine layer was not less than 2mm. According to data from the literature [9,35,36], an intermittent cutting technique produces greater cutting effectiveness and a lower overall temperature increase because heat dissipation can manifest during intervals of inactivity when the bur is not engaged. In addition, we used a high-speed handpiece equipped with a 5-hole spray for optimum cooling. However, even if manufacturers stated that greater port numbers enhance cooling efficiency, though study findings remain inconclusive. Thus, Chua et al. [28] observed that there was no statistical significance between the intra-pulpal temperatures after using high-speed air turbine featuring varied coolant port configurations(1-, 3- and 4-ports), while Lau et al. [4] found a statistically significant difference between the cooling efficiency with 1- and 4-port coolant design on electric micromotor high-speed dental handpieces. Related to the degree of wear of the burs, following the results presented in Table 4 and Figure 5, we can observe that the changes in pulpal temperature were not influenced differently by the degree of wear of the burs used. Also, under the conditions of the present study (the burs used in group C were on the eighth use, for 3 min each time) there was no heavy loading of the burs. Furthermore, in our study, a significant temperature drop was recorded for all the elements included in the sample, regardless of the condition of the pulp chamber (with pulp tissue or with thermoconductive paste). Therefore, these results, achieved within the framework of the present study, carried out following the established study design, do not support the rejection of the null hypothesis.

Consequently, the different degrees of wear of conventional diamond burs used for tangential vestibular prosthetic veneer preparation do not produce statistically significant different changes in the temperature of the pulp chamber of the teeth.

However, in the present study the working temperature was 25 °C, during the entire study, we could not ensure the temperature of the teeth in the oral cavity. Also, since it was an in vitro study, the absence of the pulp vascular microdynamics, of the periodontal tissue and the surrounded tissues from the oral cavity, can influence the temperature changes at the level of the tooth. According to Lau et al. [4], the exact quantification of the amount of heat transferred to the pulp during dental procedures is very difficult to achieve, knowing that in vivo clinical studies have many variables that require rigorous control. In addition, the density of the dentinal tubules, their orientation and structure influence the thermal conductivity of the dentin. It must also be taken into account that the thickness of the dentin layer, left after dental preparation, can influence the density of the dentinal tubules. These anatomical attributes of teeth exhibit considerable diversity, not only within a single tooth but also for different teeth [37,38,39,40,41].

The amount of heat transfer resulting from the thermal exposure at the level of the prepared dentin surface, and implicitly the risk of pulp tissue alteration, can be determined considering the thickness of the remaining dentinal layer and the thermal conductivity coefficient of the dentin. The relationship between the three variables is illustrated by a modified thermodynamic equation and shows that heat transfer through dentin is directly correlated to thermal conductivity (TC) and inversely correlated to residual dentin thickness [4,42,43].

Even so, in the in vivo activity, there are several variables that can influence the temperature evolution at the level of the dental pulp. In agreement with the relevant literature, in 2012, Thomas et al. [31], stated that two main groups of precautions must be considered when preparing a vital tooth: equipment-related precautions (apply a high-speed handpiece with an effective water-cooling system, directing the coolant toward the bur–dentin interface, using a high volume of water-cooling, with 40 mL/min preferably, and the temperature of the water-coolant must be below 35 °C) and operator-related precautions (use of light intermittent cutting strokes, avoidance of the iatrogenic removal of excessive dentin, avoid the use of dull cutting instruments, and the handpiece should be maintained in good working condition).

For all these reasons, many authors consider that the temperature values measured by in vitro studies cannot be directly applied to temperature changes in vivo [4,37,38].

The following aspects could be considered as limitations of this study:-The sample size was small; hence, further studies with a larger sample size are needed.-In the in vitro study, we could not survey the flow of dentinal fluid in the dentin tubules and the perfusion of blood or the surrounding periodontal tissues, which may have a critical impact in the temperature regulation of the dental pulp.-In this study, we could not control and standardize rigorously all the variables that can influence the temperature of the pulp chamber during the prosthetic preparation, such as the initial temperature of the teeth, which was not the same as that in the oral cavity.

## 5. Conclusions

In conclusion, under the conditions of the current study, the prosthetic veneer preparation of the frontal teeth, using conventional diamond burs with different degrees of wear (first use, fifth use, and eighth use) under water cooling, produced similar but highly statistically significant decreases in temperature at the level of the teeth pulp chamber. Thus, the degree of wear of conventional diamond burs does not influence differently the level of the temperature changes.

## Figures and Tables

**Figure 1 dentistry-11-00197-f001:**
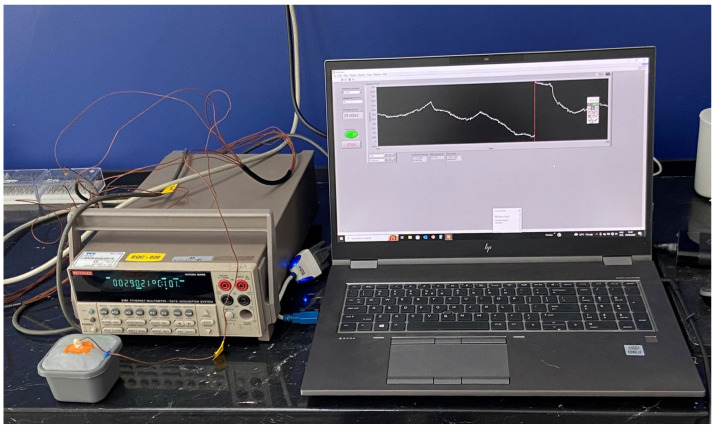
The dental pulp temperature acquisition system.

**Figure 2 dentistry-11-00197-f002:**
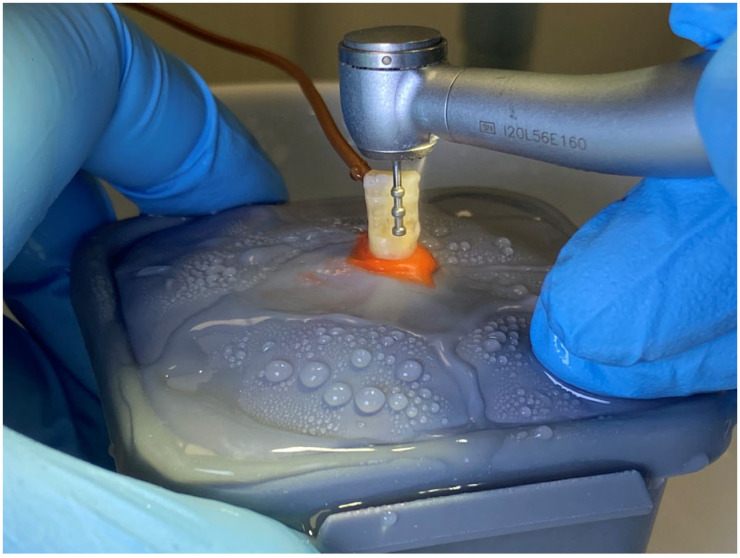
Drawing guide grooves, in axial incidence, using guide cutters.

**Figure 3 dentistry-11-00197-f003:**
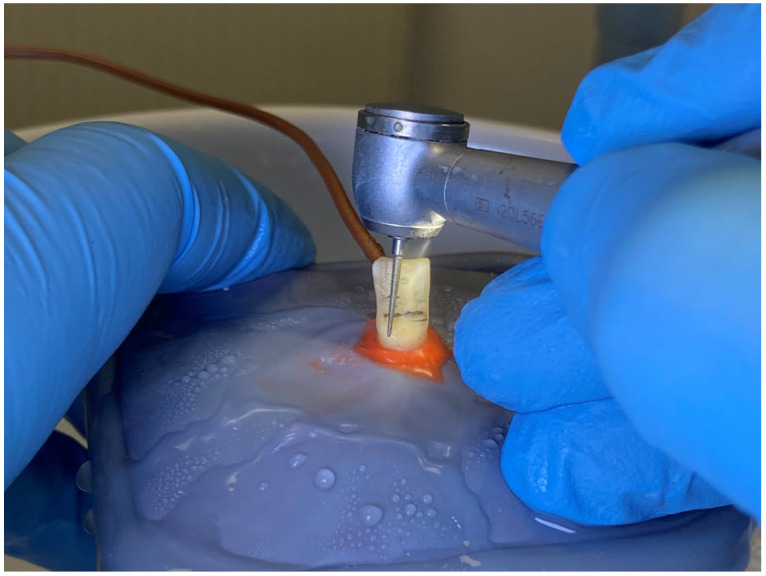
Tangential preparation of the vestibular surface, using a high-speed diamond bur and water cooling.

**Figure 4 dentistry-11-00197-f004:**
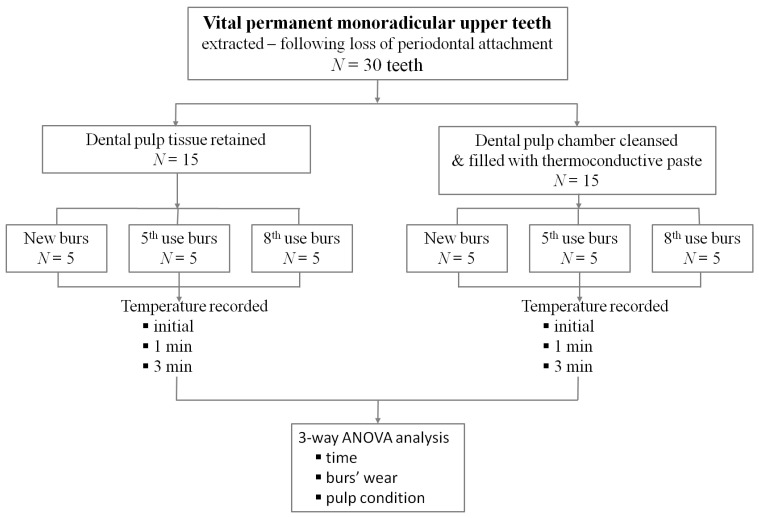
The flowchart for study protocol.

**Figure 5 dentistry-11-00197-f005:**
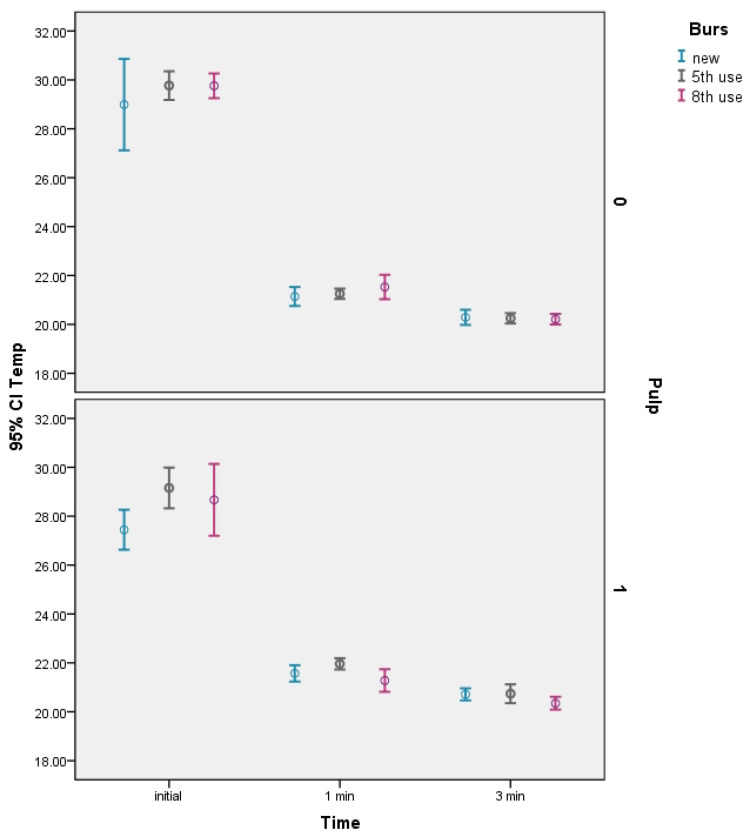
Diagram with 95% confidence intervals of the mean temperature for the 18 combinations of factors considered in the experiment.

**Table 1 dentistry-11-00197-t001:** The group distribution of the sample elements, according to the study design.

Group	Diamond Burs’ Wear	Sample Size	Pulp Chamber Condition
A	new	*N* = 10	*N* = 5 pulp tissue retained
			*N* = 5 thermoconductive paste
B	fifth use	*N* = 10	*N* = 5 pulp tissue retained
			*N* = 5 thermoconductive paste
C	eighth use	*N* = 10	*N* = 5 pulp tissue retained
			*N* = 5 thermoconductive paste

**Table 2 dentistry-11-00197-t002:** Descriptive statistics for the temperature measured in the experiment (*N* total = 90).

		Time of Temperature Measurement					
Tooth Condition	Burs’ Wear	Initial		1 min		3 min	
with pulp	new	*N* = 5	27.45 ± 0.66	*N* = 5	21.57 ± 0.27	*N* = 5	20.71 ± 0.20
	fifth use	*N* = 5	29.15 ± 0.67	*N* = 5	21.96 ± 0.19	*N* = 5	20.74 ± 0.31
	eighth use	*N* = 5	28.67 ± 1.18	*N* = 5	21.28 ± 0.37	*N* = 5	20.35 ± 0.21
with paste	new	*N* = 5	28.99 ± 1.50	*N* = 5	21.14 ± 0.31	*N* = 5	20.29 ± 0.25
	fifth use	*N* = 5	29.77 ± 0.47	*N* = 5	21.25 ± 0.17	*N* = 5	20.25 ± 0.17
	eighth use	*N* = 5	29.76 ± 0.41	*N* = 5	21.53 ± 0.40	*N* = 5	20.21 ± 0.17

Note: values are expressed as mean ± standard deviation.

**Table 3 dentistry-11-00197-t003:** ANOVA three-way analysis of the full factorial experiment data.

	Factor	F Statistic (df)	*p* Value
Main effects	Time of measurement	2030.273 (2)	<0.001 **
	Burs’ wear	5.784 (2)	0.005 **
	Pulp condition	1.503 (1)	0.224
Two-way interactions	Time * Burs	4.382 (4)	0.003 **
	Time * Pulp	15.398 (2)	<0.001 **
	Burs * Pulp	2.188 (2)	0.120
Three-way interaction	Time * Burs * Pulp	0.830 (4)	0.510

Abbreviation: df, degrees of freedom. **, *p* < 0.01 (high statistical significance).

**Table 4 dentistry-11-00197-t004:** Post hoc multiple comparisons for the temperature with the Tukey procedure.

Comparisons (Tukey Procedure)		*p* Value
Time of measurement	Initial vs. at 1 min	<0.001 **
	Initial vs. at 3 min	<0.001 **
	At 1 min vs. at 3 min	<0.001 **
Burs’ wear	New vs. fourth use	0.003 **
	New vs. eighth use	0.154
	Fourth use vs. eighth use	0.287

**, *p* < 0.01 (high statistical significance).

## Data Availability

The data can be accessed by the researchers who participated in this study and are not publicly stored on servers.

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
