# Peer review of "Evaluation of the Pulp Chamber Temperature during Tooth Veneer Preparation Using Burs with Different Degrees of Wear—A Preliminary In Vitro Study"

_dentistry, 2023, doi:10.3390/dj11080197_

Round 1
Reviewer 1 Report
Nice idea transferred to an in vitro study
Introduction : There should be more analysis in factors that are related to increase of pulp temperature. e.g. electric micromotor vs air turbine hand pieces, cutting technique (Lau XE, Liu X, Chua H, Wang WJ, Dias M, Choi JJE. Heat generated during dental treatments affecting intrapulpal temperature: a review. Clin Oral Investig. 2023;27(5):2277-2297. doi:10.1007/s00784-023-04951-1).
I cannot be sure for the limitations of this study since worn burs require more pressure by the clinician to cut. It would be more precise if you changed the water cooling flow rate.
Author Response
We would like to thank you for your appreciation of our work and say we are glad to hear that you found it interesting. We have taken into consideration your recommendations, therefore we have operated the following changes as can be seen in the attached answer:
Kind regards,
All the Authors

Reviewer 2 Report
Dear authors,
Thank you for submitting the manuscript entitled "Evaluation of the pulp chamber temperature during tooth ve-2 neer preparation using burs with different degrees of wear—A 3 preliminary in vitro study" to Dentistry Journal.
The subject is of some relevance and has importance for clinical use by generalists and specialists because the generation of heat in the tooth preparation process is important for both the patient and the clinician.
However, in the material and method, you did not indicate whether the test was performed at room or body temperature. This a bias. The temperature of the room can influence the result or even vary the results as there is no periodontal ligament or bone. In addition, you should improve the discussion that remains very brief after the results presented.
Although the authors directly address the temperature differences caused by measurement at the root surface in air or liquid media, it would be helpful to include a heat transfer equation or perhaps address the differences known for the heat transfer coefficient in air versus water. This would better illustrate the magnitude of heat transfer away from the tooth surface through surrounding fluids and soft tissues.
Best regards
Author Response
We would like to thank you for revising our article and for appreciating our work. We have taken into consideration your recommendations, therefore we have operated the following changes as can be seen in the attached answer:
Kind regards,
All the Authors

Reviewer 3 Report
In this paper, the authors have evaluated whether the different degrees of wear of the diamond burs significantly influenced the temperature changes in the pulp chamber during tangential veneer preparation.
Introduction
The introduction must be improved. From the 1960s, different studies recommended that dental pulp temperature increases should not exceed 5,5° C. The Authors in their paper have not considered and discussed the biological effects of heat on the dental pulp and the dentin.
Materials and Methods/Results
Please provide tables, illustrations, and flowcharts to clarify the whole experiment.
Discussion
Based on the references cited, the discussion of the results must be improved.
Line 36. "The Dental pulp" instead of" Dental pulp";
Line 43. "intrapulpal" instead of "intrapurpar£;
Line 44. "that" instead of "that that";
Lines 81-82. "intrapulpal" instead of"the intrapurpar";
Line 158. Please change the plural noun: "five minutes"instead of "five minute";
Line 159. Consider adding the comma before "in";
Line 160. "it is recommended not to use" instead of "it is recommended to not use";
Line 172. "filled" instead of "filed"
Line 269. Please remove the colon after "such as";
Line 277. "temperature" instead of "the temperature".
Author Response
Thank you for rigorously reviewing of our article and your detailed feedback which helps substantially in the improvement of our work. Consequently, we made the following changes which can be seen in the attached material:
Best wishes,
All the Authors

Reviewer 4 Report
Evaluation of the pulp chamber temperature during tooth veneer preparation using burs with different degrees of wear—A preliminary in vitro study
This in vitro study evaluated three conditions that may influence temperature changes in the pulp chamber during tangential veneer preparation: (a) the time of measurement during veneer preparation; (b) the degree of the diamonds burs wear; (c) the dental pulp status. The topic is very relevant, as elevation of pulp temperature during restorative therapy can lead to dental pulp pathology.
Major issues:
1. There is no description of the temperature in which the pulp-containing teeth were treated- and this is important, as such experiments should try to stimulate the mouth and pulp temperature. Several studies have highlighted the importance of conducting the experiments in a way that simulates the baseline pulp temperature, as experiments carried out at room temperature had a significant impact on the temperature profile.
2. Writing style needs improvement.
Minor issues:
Introduction
Please add references. All the first sentences of the introduction need references.
The writing should be ameliorated- the introduction should be more succinct. Use the word “the” less, its use in the manuscript is frequently inappropriate. For example, In line 65: “Zarow et al. [9] claimed that the ceramic veneers showed 65 a satisfactory clinical performance both on vital teeth (VT)…” or in line 76: “These worn-out burs may require excessive pressure application…”. Another example of the need for editing is in the sentence on line 83“The aim of this in vitro study was to evaluate whether the different degrees of wear of the diamond burs significantly differently influenced the temperature…”
Methods
This section is clear and detailed. Because the study design includes several groups, I would suggest the authors add a table that describes the groups.
There is no description of the temperature in which the pulp-containing teeth were treated- and this is important, as such experiments should try to stimulate the mouth and pulp temperature. Also, the pulp of the experimental teeth cannot be referred to as “preserved pulp”- the pulp of extracted teeth loses its viability after a few hours.
Discussion
Some of the first sentences in the discussion belong in the methods section (the statistical software details) and some belong in the results section (all of the 3rd paragraph). This happens also later on in the discussion.
The writing should be ameliorated- the introduction should be more succinct. Use the word “the” less, its use in the manuscript is frequently inappropriate. For example, In line 65: “Zarow et al. [9] claimed that the ceramic veneers showed 65 a satisfactory clinical performance both on vital teeth (VT)…” or in line 76: “These worn-out burs may require excessive pressure application…”. Another example of the need for editing is in the sentence on line 83“The aim of this in vitro study was to evaluate whether the different degrees of wear of the diamond burs significantly differently influenced the temperature…”
Author Response
We thank you for revising our article and for you valuable suggestions. In what follows we have taken into consideration your guidelines as can be seen in the attached answer.
All the best,
All the Authors

Round 2
Reviewer 3 Report
COMMENTS ON THE QUALITY OF ENGLISH LANGUAGE
Line 85. Please correct your spelling. The word “hand pieces” seems to be miswritten. Consider replacing it.
“handpieces” instead of “hand pieces”;
Line 230. Please consider inserting a comma before “and the dental pulp status”;
Line 249. Please consider inserting a comma before “and Laforgia et al.”;
Line 257. Please correct article usage
“heat” instead of “a heat”;
Line 258. Please correct article usage
“periods of rest” instead of “the periods of rest”;
Line 259. Please correct your spelling. The word “handpice” seems to be miswritten. Consider replacing it.
Line 259. Please correct article usage:
“with a 5 hole-spray” instead of “with 5 hole-sprey”;
Line 261. Please correct your spelling. The word “ambigous” seems to be miswritten. Consider replacing it.
“ambiguous” instead of “ambigous”;
Line 263. It appears that “actually” may be unnecessary in this sentence. Please consider removing it.
Line 256. Please consider inserting a comma before “an intermittent cutting technique”;
Line 270. Please correct article usage
“a significant” instead of “the significant”;
Line 274. Please correct article usage
“the rejection” instead of “rejection”;
Line 282. “and the surrounding tissues” instead of “and of the surrounded tissues”;
Line 300. Please correct your spelling. The word “directingthe” seems to be miswritten. Consider replacing it.
Line 302. “preferably” instead of “preferrably”.
COMMENTS ON THE QUALITY OF ENGLISH LANGUAGE
Line 85. Please correct your spelling. The word “hand pieces” seems to be miswritten. Consider replacing it.
“handpieces” instead of “hand pieces”;
Line 230. Please consider inserting a comma before “and the dental pulp status”;
Line 249. Please consider inserting a comma before “and Laforgia et al.”;
Line 257. Please correct article usage
“heat” instead of “a heat”;
Line 258. Please correct article usage
“periods of rest” instead of “the periods of rest”;
Line 259. Please correct your spelling. The word “handpice” seems to be miswritten. Consider replacing it.
Line 259. Please correct article usage:
“with a 5 hole-spray” instead of “with 5 hole-sprey”;
Line 261. Please correct your spelling. The word “ambigous” seems to be miswritten. Consider replacing it.
“ambiguous” instead of “ambigous”;
Line 263. It appears that “actually” may be unnecessary in this sentence. Please consider removing it.
Line 256. Please consider inserting a comma before “an intermittent cutting technique”;
Line 270. Please correct article usage
“a significant” instead of “the significant”;
Line 274. Please correct article usage
“the rejection” instead of “rejection”;
Line 282. “and the surrounding tissues” instead of “and of the surrounded tissues”;
Line 300. Please correct your spelling. The word “directingthe” seems to be miswritten. Consider replacing it.
Line 302. “preferably” instead of “preferrably”.
Author Response
Dear Reviewer,
Thank you for your valuable comments and suggestions regarding the issues pertaining to the quality of the English language; we took them into consideration and made the necessary alterations.
Best regards,
All the Authors
